# Scots Pine (*Pinus sylvestris* L.) Ecotypes Response to Accumulation of Heavy Metals during Reforestation on Chalk Outcrops

Vladimir M. Kosolapov [1], Vladmir I. Cherniavskih [1], Elena V. Dumacheva [1], Luiza D. Sajfutdinova [1], Alexey A. Zavalin [2], Alexey P. Glinushkin [3], Valentina G. Kosolapova [4], Bakhyt B. Kartabaeva [5], Inna V. Zamulina [6], Valery P. Kalinitchenko [5,7,*], Michail G. Baryshev [5], Michail A. Sevostyanov [5,8], Larisa L. Sviridova [5], Victor A. Chaplygin [6], Leonid V. Perelomov [9], Saglara S. Mandzhieva [6], Marina V. Burachevskaya [9] and Lenar R. Valiullin [5,10,11]

1 Federal Williams Research Center of Forage Production and Agroecology, 1 Nauczny Gorodok, 141055 Lobnya, Russia; kormoproizvodstvo@yandex.ru (V.M.K.); cherniavskih@mail.ru (V.I.C.); dumacheva63@mail.ru (E.V.D.); louisa_45@mail.ru (L.D.S.)

2 All-Russian Research Institute for Agrochemistry Named after D.N. Pryanishnikov of the Russian Academy of Sciences, 31a, Pryanishnikova St., 127434 Moscow, Russia; otdzem@mail.ru

3 Russian Academy of Sciences, 32-a Gagarinsky, Leninsky Ave., 119991 Moscow, Russia; glinale1@mail.ru

4 Moscow Timiryazev Agricultural Academy, 49 Timiryazevskaya St., 127422 Moscow, Russia; v.kosolapova@rgau-msha.ru

5 All-Russian Research Institute of Phytopathology, 5 Ownership, Institute St., Odintsovo District, 143050 Big Vyazemy, Russia; kartabaeva040893@mail.ru (B.B.K.); baryshev_mg@mail.ru (M.G.B.); cmakp@mail.ru (M.A.S.); sviridovalarisal@rambler.ru (L.L.S.); valiullin27@mail.ru (L.R.V.)

6 Academy of Biology and Biotechnology, Southern Federal University, 105/42 Bolshaya Sadovaya St., 344006 Rostov-on-Don, Russia; inir82@mail.ru (I.V.Z.); otshelnic87.ru@mail.ru (V.A.C.); msaglara@mail.ru (S.S.M.)

7 Institute of Fertility of Soils of South Russia, 346493 Persianovka, Russia

8 Institute of Metallurgy and Materials Science Named after A.A. Baikov, 119334 Moscow, Russia

9 Laboratory of Soil Chemistry and Ecology, Tula State Lev Tolstoy Pedagogical University, 125 Lenin Ave., 300026 Tula, Russia; perelomov@rambler.ru (L.V.P.); marina.0911@mail.ru (M.V.B.)

10 Federal Center for Toxicological, Radiation and Biological Safety, 420075 Kazan, Russia

11 Faculty of Forestry and Ecology, Kazan State Agrarian University, 420015 Kazan, Russia

* Correspondence: kalinitch@mail.ru; Tel.: +7-(918)5333041

**Abstract:** As objects for reforestation, the least studied are carbonate substrates, which have a number of specific features in terms of mineral composition, the exchange of nutrients, and biological activity. The use of biological preparations of a consortium of bacteria of the genus *Bacillus* and mycorrhizal fungi of the genus *Glomus* in growing seedlings of Scots pine (*Pinus sylvestris* L.) on carbonate substrates provides the metabolic products; soluble and microelement salts function as catalysts for chemical reactions of exudates and soil products; and a greater amount of plant heavy metals (HM) Cu, Zn, Cd, and Pb accumulate in the soil. Among HMs, the random factors most strongly determined an accumulation of Cd (the influence rate of random factors $h^2_x = 34.6\%$) and Pb (the influence rate of random factors $h^2_x = 21.7\%$) in the plants. A trend of all studied HMs higher uptake by the Cretaceous pine (*Pinus sylvestris* var. *cretacea* (Kalen.) Kom.) in comparison with the *P. sylvestris* ecotype is revealed. Against the biological preparation background of Biogor KM and MycoCrop®, a greater value of the HM's biological absorption in comparison with the option without biological preparations is noted. This process occurs against a background of a significant increase in the nitrification capacity in the chalk fine-grained substrate (soil aggregates < 1 mm in size), which is an indirect indicator of an increased intensity of microbiological processes. Spearman's correlation was noted between the coefficient of accumulation of Cu, Zn, Cd, and Pb in the dry matter of Scots pine (*P. sylvestris*) seedlings and the nitrification capacity of substrate ($r_s = 0.610$–$0.744$, $p < 0.05$), as well as the relationship between the nitrification capacity index of substrate and the coefficient of biological absorption of copper, zinc, and cadmium ($r_s = 0.543$–$0.765$, $p < 0.05$). No relationship was found between the coefficient of biological absorption of lead and other soil chemical property

indicators. HM absorption by plants was random. No correlations have been established between an accumulation of HMs and a content of total nitrogen, an absolute value of nitrate nitrogen, a humus content, or a pH. The significance of the work is the possibility of providing reliable reforestation with Scots pine (*P. sylvestris*) and Cretaceous pine (*P. sylvestris* var. *cretacea*) on the chalk outcrops using the biological preparations Biogor KM, MycoCrop®, and BGT* methodology and ensuring soil phytoremediation from HMs.

**Keywords:** *Pinus sylvestris* L.; *Pinus sylvestris* var. *cretacea* (Kalen.) Kom.; degraded soil on chalk outcrops; heavy metals; biological product Biogor KM; Biogeosystem Technique

## 1. Introduction

The basis of economic, environmental, and political activity in various regions of the world is low-productive lands unsuitable for intensive agriculture and forestry development [1–5]. The need for both forest reclamation measures and projects for reforestation and reintroduction of forest plantations in areas of growth in the past is being actively discussed [6–9]. The natural mechanisms of absorption, biotransformation, and bioaccumulation of pollutants in plants are to be accounted for.

The creation of forest plantations on degraded lands is considered a reserve for sustainable development and an effective mechanism for increasing biodiversity, stabilizing soil fertility, sequestering $CO_2$, and reducing the harmful chemicals and heavy metals (HMs) rate of transfer in ecosystems [10–14].

For a comprehensive assessment of the degraded soils of the world and the development of reforestation techniques, research is needed in regions with the most severe forest growth conditions. A soil substrate is of great importance as the basis of reforestation measures. In this regard, providing forest plants with minerals in conditions of soil fertility deficiency is of particular importance.

With insufficient nutrition, characteristic of soils with low fertility, some elements, even at a minimal amount, can have a toxic effect. It is important to study the processes of absorption of HMs on specific substrates: saline soils, various kinds of outcrops of soil-forming rocks, and dumps of industrial developments with low agrochemical and biological activity. Of particular interest are the questions of the interaction of the woody plants with the soil microflora under these conditions.

As objects for reforestation, the least studied are carbonate substrates, which have a number of specific features in terms of mineral composition, the exchange of nutrients, and biological activity [15,16].

The south of the Central Russian Upland has a number of features associated with a wide distribution of ravine-gully complexes formed on the carbonate soil-forming rocks. The landscape has a low level of projective cover. It is distinguished by a high level of vertical and lateral mobility of substrates associated with a cryogenic process, a constant exposure to water runoff, and an intense process of geological weathering. The features are significant annual and average daily temperature fluctuations and a high albedo of the soil surface [15–18].

Of particular importance is the rockiness of chalk substrates with a high proportion of "skeletal" part (soil particles > 1 mm) compared to "fine grained aggregates" (soil particles < 1 mm). In this regard, previous studies have shown that local soil formation occurs on these substrates. A biological process does not occur in the entire volume of the substrate but in cracks between large soil units in which small particles < 1 mm in size are located. In these cracks, the accumulation and decomposition of plant residues, the accumulation of various forms of nitrogen, and humidification run [19–22].

Figure 1 shows the profile of soddy-calcareous soil on the eluvium-chalk parent rock in the ravine-gully complex.

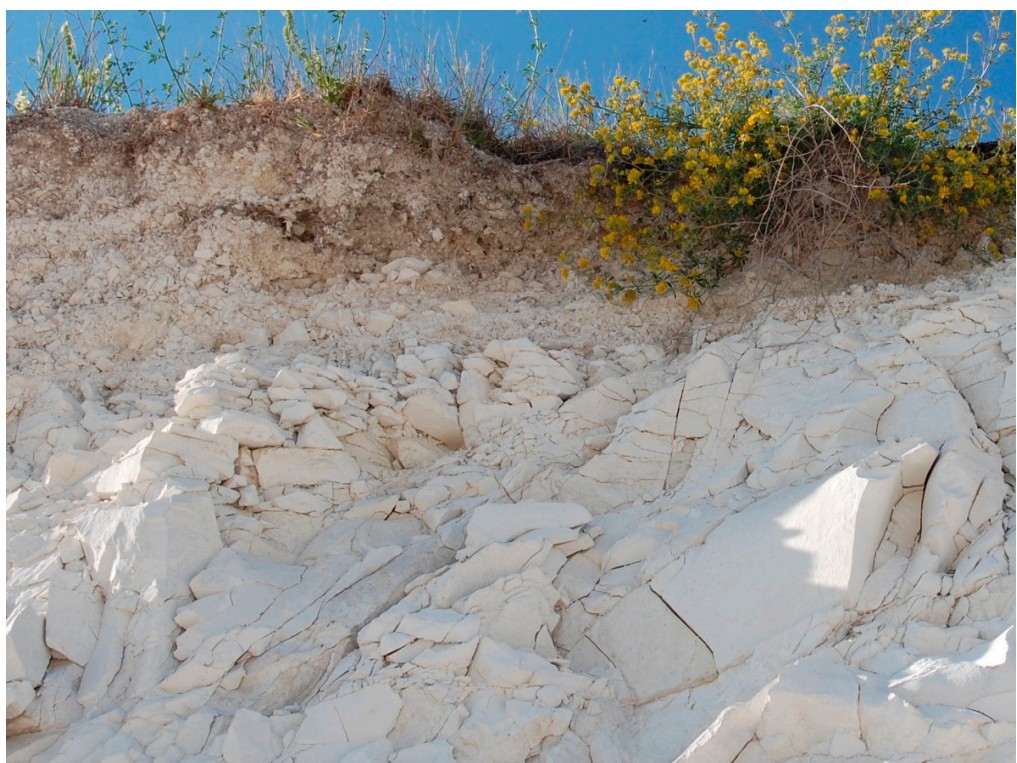

**Figure 1.** Soddy-calcareous soil profile on the eluvium chalk parent rock. Example of a soil profile for ravine-gully complexes with chalk outcrops (Vatutino village, Valuysky district, Belgorod region. Photo by V.I. Chernyavskih).

The specificity of the substrate determines the endemism and specificity of plants capable of growing under poor development conditions [23–25]. In this regard, the forest reclamation measures require a special approach in the choice of forest crops, cultivation technologies, and approaches to the study of factors affecting the growth and development of forest crops, especially in the early stages of development.

Pine is an obligate mycotroph capable of entering into symbiosis with 200–300 species of ectomycorrhizal fungi. A host plant, providing fungi with photosynthesis products, is able to assimilate water and mineral nutrients due to ectomycorrhiza, and mycorrhiza increases plant resistance to adverse environmental factors.

The most valuable forest culture is *P. sylvestris*—Scots pine, which has the widest ecological amplitude and a large number of ecotypes [26–28]. Scots pine is of great importance for degraded landscapes due to a number of features and properties, such as high polymorphism and high ecological amplitude. *P. sylvestris* is successfully growing in various conditions. The pine culture is unique.

Of particular interest is the Cretaceous pine culture (*P. sylvestris* var. *cretacea*), an ecotype formed on Cretaceous outcrops, which is a tertiary relic of the south of the Central Russian Upland, preserved in the non-glacial zone during the last glaciation [29–33]. Previously, the effectiveness of pine cultures in the creation of forest plantations on chalk outcrops was shown [34–36]. However, the issue of the seed reproduction of Scots pine (*P. sylvestris*) ecotypes in culture and during self-recovery on chalk outcrops has not been sufficiently studied [37–39]. Particular attention should be paid to the regulation of tree growth and development processes at the initial stages of ontogenesis, starting from seed germination in connection with the nitrogen regime and humus state, closely related to the aggregation and microelement composition of the substrate.

The accumulation of toxic HMs in ecosystems and in the soil and their transport by watercourses are considered serious environmental problems that can have severe negative consequences for plants, animals, and humans. A number of microelements,

known as HMs, can have an extremely negative impact on plants and associated consortia of microorganisms, contributing to the deterioration of a soil's nutrient regime [40–42].

HMs can carry an environmental hazard [43,44]. HMs transformation, absorption by plants of various ecotypes, species, and families, and the influence of nitrification inhibitors, mineral fertilizers, soil microorganisms, ectomycorrhizal fungi, etc. are studied on various soil types [45–49].

As a geographic region, the south of the Central Russian Upland has a number of features that require special solutions to environmental problems associated with the presence of harmful substances and HMs in the ecosystem. A close occurrence of iron ore seams, agriculture, a spread of linear erosion, and a high dissection of the territory can lead to the danger of HM accumulation in agricultural landscapes and their transfer through the hydrographic network to water bodies and further along food chains. In this regard, the formation of biological barriers is necessary to block the accumulation of HM in vegetation in degraded areas with ongoing active erosion processes. Despite the fact that the carbonate substrate is a powerful solid-phase barrier for HM [40], the described local soil evolution features and a high level of local biochemical processes can somewhat change the known laws of HM uptake by plants. A well-known mycotrophy of *Pinus* species significantly enhances the ability of plants to absorb hard-to-reach elements from substrate [50–52].

In this regard, the use of biological preparations based on fungi and bacteria, as well as their consortiums and associations with biologically active substances, makes it possible to create technologies for the effective development of both herbaceous and woody vegetation on low-productive lands [53–55]. It has been shown that biopreparations based on a consortium of fungi from the order *Glomales* can increase the germination rate of seeds of various *Pinus* species [56,57], as well as the survival rate of young plants [58–61].

However, an issue of HM uptake by *P. sylvestris* plants raises concern [62–64]. It has been shown that HMs have a variety of negative effects on cytogenetic and biochemical features and seed germination of *P. sylvestris* [65–67]. Some researchers consider *P. sylvestris* a model plant for studying a coniferous plant's adaptation to the action of HMs [68].

The reforestation of the Scots pine (*P. sylvestris*) and Cretaceous pine (*P. sylvestris* var. *cretacea*) on the chalk outcrops using the biological preparations Biogor KM and MycoCrop® is promising for soil phytoremediation from HMs.

The aim of the research was to assess HMs accumulation and biological absorption in the seedlings of two ecotypes of *P. sylvestris* depending on the application of biopreparations based on a consortium of fungi from the order *Glomales* and bacteria of the genus *Bacillus*. Biologically active substances, their metabolic products, and microelements were studied during reforestation on the chalk outcrops. Reforestation with Scots pine and Cretaceous pine on the chalk outcrops can ensure soil phytoremediation from HMs. Prospects for improvement of soil environment services via the Biogeosystem Technique (BGT*) methodology have been taken into account [69].

## 2. Materials and Methods

### 2.1. Research Area

The studies were carried out in the conditions of erosional landscapes in the south of the Central Russian Upland, in ravine-gully complexes with chalk outcrops. Field experiments were carried out on the territory of the Belgorod region (Russia). The region is largely subject to water erosion, the spread of ravines, and gullies, with a territory dissection coefficient of 1.4–1.7 km km$^{-2}$. A feature of the region is the spread of the iron ore and steel industries, with the extraction of iron ore by open and closed methods, intensive agricultural production, and transport. The region is characterized by a temperate continental climate with an average annual air temperature of 5.4 °C to 6.7 °C and an average annual rainfall of 465–550 mm, including 60–75 mm of precipitation during the growing season.

Two ecotypes of a Scots pine were taken for the study: Scots pine (*P. sylvestris*), the standard ecotype, and Cretaceous pine (*Pinus sylvestris* var. *cretacea*), a tertiary relic,

which is considered a separate species and referred to as the ecotype *Pinus sylvestris* var. *cretacea* [29,30,32]. Seeds for the experiment were selected from typical habitats in the Belgorod region.

### 2.2. Object Conditions and Experiment Layout

Field studies were carried out on a chalk outcrop (50.452995° N; 37.736746° E) near the Verkhniye Lubyanki village in the Volokonovsky district of Belgorod oblast on a southwestern slope. The experimental plot was chosen on the left bank of the Oskol River tributary. A soil substrate is an outcrop of eluvium chalk (Figure 2).

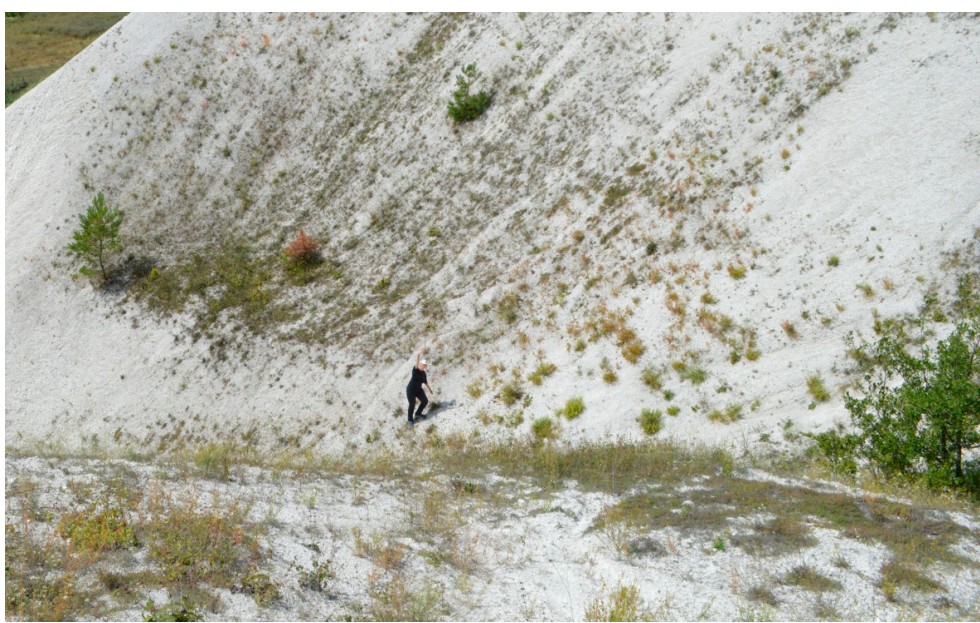

**Figure 2.** Area of experiment carry out (photo by V.I. Chernyavskih).

Weather conditions during the research period in 2018–2020 were characterized by different amounts of precipitation (429–693 mm, 77.5–125.3% of the average annual norm) and elevated average annual temperatures (9.2–10.4 °C, 2.9–4.1 °C above the average annual norm). A value of Selyaninov hydrothermal moisture coefficient (HTC) K = R × 10/Σt (where R is the sum of precipitation, mm, for the period with temperatures above +10 °C, Σt is the sum of temperatures above +10 °C) ranged from 0.59 to 1.22 for the same period.

In a two-factor field experiment, we studied the accumulation of HMs in the vegetative mass of seedlings of two Scots pine ecotypes: *P. sylvestris* and *P. sylvestris* var. *cretacea*, depending on the seeds treatment with a biological product based on a consortium of microorganisms and biologically active substances during sowing.

The general scheme of experiment is shown in Table 1.

**Table 1.** Research design (2018–2020).

| Pine Ecotype (Factor A) | | Biological Product (Factor B) | |
|---|---|---|---|
| A1 | *Pinus sylvestris* L. | B1 | distilled water (c) |
| | | B2 | Biogor KM |
| | | B3 | MycoCrop® |
| A2 | *Pinus sylvestris* var. *cretacea* (Kalen.) Kom. | B1 | distilled water (c) |
| | | B2 | Biogor KM |
| | | B3 | MycoCrop® |

The experiment was performed in triplicate.

In each replicate plot of the experiment, seeds were sown at a rate of 100 pcs. viable seeds per 1 m$^2$. Thus, 200 seeds were sown per registration plot (2 m$^2$), and 500 seeds were sown per total plot (5 m$^2$). In the spring of 2019, the number of seedlings, depending on the options for the experiment, varied from 41.5 to 55.8 pcs per 1 m$^2$ [70].

An accounting plot area of 2 m$^2$ (1 m $\times$ 2 m) has been chosen as optimal for laboratory and field experiments with coniferous crop seedlings based on previous studies [69]. The total area of the replicate plot was 5 m$^2$ (2.5 m $\times$ 2 m). The total area of the gross experiment was 100 m$^2$. The total area of the experimental plots was 90 m$^2$. The total net accounting area of the plots was 36 m$^2$.

By the autumn of 2020, depending on the variant of the experiment, the number of seedlings in each study plot ranged from 11.3 to 27.8 pcs m$^{-2}$.

A mixed sample for determining HM content was formed from 20 biennial seedlings for each repetition of the experiment in triplicate.

The studied biopreparations, Biogor KM (Russia) and MycoCrop$^{\circledR}$, are available on the market.

Biogor KM (Russia) is a biological preparation on a liquid carrier based on a consortium of bacteria of the genus *Bacillus* and mycorrhizal fungi of the genus *Glomus*. Metabolic products, soluble salts, and salts of microelements act as catalysts for chemical reactions in exudates and products. The composition includes six strains of microorganisms. The method of application is a finely dispersed spraying of seeds before sowing.

MycoCrop$^{\circledR}$ (Germany) is a preparation based on the fungi *Glomus proliferum*, *G. intradice*, *G. etunicatum*, and *G. mosseae*. A carrier is clay microgranules. The method of application is an introduction into the soil together with the seeds during sowing.

### 2.3. Sampling and Research

Soil samples were taken before sowing at a depth of 20 cm with a drill using the envelope method in five places in each plot, and a mixed sample in each plot was formed for analysis. After preparing the mixed sample, the soil was brought to an air-dry state. By sieving with a round cell d = 1 mm, the substrate was mechanically separated to the size fraction less than 1 mm and to the skeletal part with a size of more than 1 mm, not affected by the soil-forming process.

For the analysis, only the fine aggregate fraction of less than 1 mm, which accumulates in cracks between large soil units and plays a fundamental role in the growth and development of plants on carbonate outcrops, was used. A fine aggregate fraction of less than 1 mm was ground and used for further analysis.

The humus content in the soil was determined according to the Tyurin method in the modification of TsINAO (Soils. Methods for determination of organic matter). For oxidation, a solution of potassium dichromate in sulfuric acid was used. The trivalent chromium equivalent to the content of organic matter was determined on a photoelectric colorimeter. The photometry of the solutions was carried out in a cuvette with a transillumination thickness of 2 cm relative to the reference solution, wavelength 590 nm [71].

The determination of the pH of the dense residue and aqueous extract was carried out by the electrical conductivity method (Soils. Methods for determination of specific electrical conductivity, pH, and solid residue of water extract). The extraction of water-soluble salts from the soil was carried out with distilled water at a ratio of soil to water of 1:5, and the pH was measured using a pH meter. The pH meter was adjusted using three buffer solutions with pH 4.01, 6.86, and 9.18 prepared from standard titers [72].

The content of total nitrogen in the soil substrate was determined by photoelectrocolorimetry (Soils. Methods for determination of total nitrogen). An optical density of the solution was measured at a wavelength of 655 nm relative to a zero solution in a cuvette with an absorbing layer thickness of 1 cm. Based on the results, a calibration graph was built [73].

The content of the nitrate nitrogen was measured by the ionometric method (Soils. Determination of nitrates by the ionometric method). Nitrates were extracted with a solution of potassium alum with a mass fraction of 1% at a ratio of 1:2.5 between the mass of the soil sample and the volume of the solution. The content of nitrates in the extract was determined using an ion-selective electrode [74].

The nitrification ability of the substrate was determined by the Koravkov method. The content of nitrates in the soil was measured by the ionometric method according to the methodology described above. Then, the soil substrate was composted in a thermostat at a temperature of 28 °C and a humidity of 60% for 7 days. The content of the nitrates was determined again. A difference between the initial content of nitrates in the substrate and the content after composting is an indicator of nitrification ability. That is, an ability of the substrate to accumulate nitrates under the influence of microbiological processes [75].

The content of HMs in soil and plant samples was determined in accordance with the guidelines for the determination of HMs in soils of agricultural lands and products [76].

The Cu, Zn, Cd, and Pb extractable (mobile) forms in the soil were determined by atomic absorption spectroscopy with flame atomization. An extraction was carried out from the separate samples in two repetitions. An ammonium acetate buffer solution (AAB) with pH 4.8 was used as an extractant. A single soil sample mass was 10 g. The ratio of soil to solution was 1:10. Used analytical lines: for Zn—213.8 nm, for Cu—324.7 nm, Pb—217.0 nm, and for Cd—228.8 nm. The gas mixture was acetylene–air [76].

Biennial seedlings were used to determine HMs in plants. Plants were uprooted, brought to an air-dry state, transferred to laboratory conditions, and analyzed in accordance with the methods.

The HMs, Cu, Zn, Cd, and Pb content in plant samples was determined in an ash solution on an atomic absorption spectrophotometer. The plant samples were mineralized via dry ashing [77]. An acid extraction of a metal from the ash was carried out with diluted nitric acid (1:1). The extract was kept in a boiling water bath for 30 min. The settled filtrate was used for analysis. The determination of a metal in the ash solution was carried out on an atomic absorption spectrophotometer with flame atomization according to the method described above for the soil.

The biological absorption coefficient (BAC) was calculated as a ratio of a mobile form of metal in the soil to its accumulation in the dry matter (DM) of plants. The whole batch of two-year-old plant seedlings was used for analysis.

### 2.4. Data Processing and Statistical Analysis

The experimental data were statistically processed. An assessment of the reliability of the results was fulfilled. A strength in the influence of organized factors was identified. The method of analysis of two-factor complex variance (ANOVA) was used. The average values (M), standard errors (m), and coefficients of variation ($C_v$, %) were calculated. Spearman's rank correlation ($r_s$) was used to identify the closeness of a relationship between the studied traits. Calculations were carried out using Microsoft Excel 10.

## 3. Results

*Accumulation of HMs in the Dry Mass of Seedlings of Two P. sylvestris Ecotypes*

It should be noted that there is a slight variation in the indicators of the chemical composition of the soil substrate. The coefficient of variation ranged from 1.43 to 9.85%, which indicates the evenness of the area where the test was carried out.

Studies have not established a significant difference in the content of humus, various forms of nitrogen, pH, or mobile forms of HMs in particles of soil substrate < 1 mm over the period of study in different variants of the experiment. Data on the analysis of soil substrate on average for 2019–2020 are shown in Table 2.

**Table 2.** Chemical composition (mean $\pm$ error) of mechanical substratum particles less than 1 mm in size in the chalk outcrop (2019–2020).

| Parameter | $M \pm m$ | Cv | lim |
|---|---|---|---|
| Mobile form Cu, mg kg$^{-1}$ | $8.12 \pm 0.54$ | 7.56 | 7.00–9.10 |
| Mobile form Zn, mg kg$^{-1}$ | $4.28 \pm 0.27$ | 7.90 | 3.73–4.90 |
| Mobile form Cd, mg kg$^{-1}$ | $0.11 \pm 0.01$ | 8.25 | 0.09–0.12 |
| Mobile form Pb, mg kg$^{-1}$ | $1.28 \pm 0.10$ | 9.85 | 1.11–1.50 |
| $N_{total}$ content, % | $0.16 \pm 0.01$ | 7.82 | 0.15–0.18 |
| N-NO$_3$ Ccontent, mg kg$^{-1}$ | $14.46 \pm 0.91$ | 7.44 | 12.90–16.30 |
| Nitrification capacity, mg kg$^{-1}$ | $17.92 \pm 1.11$ | 7.44 | 12.90–16.30 |
| Humus content, % | $2.13 \pm 0.12$ | 7.28 | 1.90–2.43 |
| pH value | $7.84 \pm 0.09$ | 1.43 | 7.66–8.03 |

Note: M is the average value; m is the mean error; Cv is the coefficient of variation; lim are limiting values of variation; n = 3.

HMs accumulation in the dry mass of seedlings against nutrition backgrounds is shown in Table 3. It was found that the differences in the absolute values of HM content were insignificant in the different ecotypes of *P. sylvestris*. The use of biological preparations provides a significant increase in the content of HM in the dry mass of seedlings in both pine ecotypes by 1.2–1.5 times. The value depended on the origin of the chemical element. The absolute content of Cu increased most significantly.

**Table 3.** Accumulation of HMs in the dry mass of seedlings of two ecotypes of *P. sylvestris* L. using biological preparations Biogor KM and MycoCrop® (mean $\pm$ error).

| Pine Ecotype | Biological Preparation | Accumulation of HMs, mg kg$^{-1}$ | | | |
|---|---|---|---|---|---|
| | | Cu | Zn | Cd | Pb |
| *Pinus sylvestris* L. | Water (c) | $3.11 \pm 0.03$ A a | $9.38 \pm 0.28$ B a | $0.11 \pm 0.01$ C a | $0.95 \pm 0.05$ B a |
| | Biogor KM | $4.58 \pm 0.10$ A b | $12.27 \pm 0.78$ B b | $0.17 \pm 0.03$ C b | $1.13 \pm 0.03$ D b |
| | MycoCrop® | $4.67 \pm 0.10$ A b | $12.55 \pm 0.31$ B b | $0.14 \pm 0.01$ C b | $1.14 \pm 0.04$ D b |
| *Pinus sylvestris* var. *cretacea* (Kalen.) Kom. | Water (c) | $3.24 \pm 0.06$ A a | $10.11 \pm 0.14$ B a | $0.12 \pm 0.01$ C a | $0.99 \pm 0.01$ D a |
| | Biogor KM | $4.79 \pm 0.11$ A b | $13.57 \pm 0.11$ B b | $0.17 \pm 0.03$ C b | $1.07 \pm 0.05$ D a |
| | MycoCrop® | $4.54 \pm 0.14$ A b | $13.31 \pm 0.12$ B b | $0.16 \pm 0.01$ C b | $1.07 \pm 0.06$ D a |

Note: In every cell of the table are presented the average value (first digit) and mean error (second digit). Different capital letters describe statistically significant differences between HMs (in rows). Different lowercase letters describe statistically significant differences between biological preparations Biogor KM and MycoCrop® (in column). Tukey HSD test, $p < 0.05$; n = 3.

A two-factor analysis of variance (ANOVA) was used to assess the influence of the studied factors "pine ecotype" and "biological product" on the resulting trait "accumulation of HMs in plants". The results are shown in Table 4. The biological preparations Biogor KM and MycoCrop® most strongly affect the content of HMs in the dry mass of pine seedlings. A significant difference between the ecotypes of *P. sylvestris* was established only in the content of Zn in the dry mass. Among the HMs, the most strongly random factors determined the accumulation of Cd (the strength of influence of random factors $h^2_x = 34.6\%$) and Pb (the strength of influence of random factors $h^2_x = 21.7\%$) in plants.

**Table 4.** Results of two-way analysis of variance (ANOVA) of the influence of various factors on the accumulation of HMs in seedlings of two ecotypes of *P. sylvestris* L. on chalk outcrops.

| Resulting Trait | Source of Variation | D | n − 1 | $s^2$ | $F_f$ | $F_{0.05}$ | $h^2_x$ |
|---|---|---|---|---|---|---|---|
| Accumulation of Cu in plants, mg kg$^{-1}$ DM | Total | 8.98 | 17 | | | | 100.0 |
| | Reps | 0.03 | 2 | | | | 0.4 |
| | Random | 0.19 | 10 | 0.02 | | | 2.1 |
| | A | 0.02 | 1 | 0.02 | 1.2 | 5 | 0.3 |
| | B | 8.63 | 2 | 4.32 | 224.0 * | 4.1 | 96.1 |
| | A × B | 0.10 | 2 | 0.05 | 2.5 | 4.1 | 1.1 |
| Accumulation of Zn in plants, mg kg$^{-1}$ DM | Total | 48.20 | 17 | | | | 100.0 |
| | Reps | 1.34 | 2 | | | | 2.8 |
| | Random | 2.25 | 10 | 0.22 | | | 4.7 |
| | A | 3.88 | 1 | 3.88 | 17.3 * | 5 | 8.1 |
| | B | 40.43 | 2 | 20.21 | 89.9 * | 4.1 | 83.9 |
| | A × B | 0.30 | 2 | 0.15 | 0.7 | 8.8 | 0.6 |
| Accumulation of Cd in plants, mg kg$^{-1}$ DM | Total | 0.019 | 17 | | | | 100.0 |
| | Reps | 0.001 | 2 | | | | 7.0 |
| | Random | 0.007 | 10 | 0.001 | | | 34.6 |
| | A | 0.001 | 1 | 0.001 | 0.9 | 8.8 | 3.2 |
| | B | 0.010 | 2 | 0.005 | 7.8 * | 4.1 | 54.2 |
| | A × B | 0.000 | 2 | 0.000 | 0.1 | 8.8 | 0.9 |
| Accumulation of Pb in plants, mg kg$^{-1}$ DM | Total | 0.129 | 17 | | | | 100.0 |
| | Reps | 0.010 | 2 | | | | 7.9 |
| | Random | 0.028 | 10 | 0.003 | | | 21.7 |
| | A | 0.004 | 1 | 0.004 | 1.6 | 5 | 3.4 |
| | B | 0.074 | 2 | 0.037 | 13.2 * | 4.1 | 57.2 |
| | A × B | 0.013 | 2 | 0.006 | 2.3 | 4.1 | 9.8 |

Note. Factor A—"pine ecotype"; factor B—"biological product"; D is the sum of squared deviations (deviant); $s^2$—dispersion; n − 1 is the number of degrees of freedom; $h^2_x$—the strength of influence on the effective attribute; *—statistically significant differences; n = 3.

Table 5 shows the biological absorption coefficient (BAC) for TMs. It was found that BAC increased against the background of the biological preparations Biogor KM and MycoCrop® in both studied ecotypes of *P. sylvestris*.

A significant increase in BAC was found for Cu in the ecotype *P. sylvestris* against the background of biological preparations Biogor KM and MycoCrop® by 1.29–1.30 times, in the ecotype *P. sylvestris* var. *cretacea* by 1.49–1.55 times; BAC Zn increased in the dry matter of *P. sylvestris* against the background of biological preparations Biogor KM and MycoCrop® by 1.3 times, and in *P. sylvestris* var. *cretacea* by 1.39–1.41 times.

The trend of higher uptake of all studied HMs by the *P. sylvestris* var. *cretacea* ecotype compared to the *P. sylvestris* ecotype was established against the background of the use of biological preparations Biogor KM and MycoCrop®. Under comparable conditions of a field experiment, when using biological preparations Biogor KM and MycoCrop®, a significant excess of BAC in *P. sylvestris* var. *cretacea* plants compared to *P. sylvestris* was found only for Cu.

The results of a two-factor analysis of variance for HMs and an assessment of the strength of the influence of factors are shown in Table 6.

**Table 5.** BAC of HMs uptake by seedlings of two *P. sylvestris* L. ecotypes under different biological preparations, Biogor KM and MycoCrop®, in the chalk substrate (mean ± error).

| Pine Ecotype | Biological Preparation | HM | | | |
|---|---|---|---|---|---|
| | | Cu | Zn | Cd | Pb |
| *Pinus sylvestris* L. | Water (c) | 0.41 ± 0.03 A a | 2.26 ± 0.07 B a | 1.02 ± 0.13 C a | 0.80 ± 0.12 C a |
| | Biogor KM | 0.54 ± 0.02 A b | 2.95 ± 0.26 B b | 1.56 ± 0.22 C b | 0.93 ± 0.04 D b |
| | MycoCrop® | 0.53 ± 0.02 A b | 3.02 ± 0.19 B b | 1.26 ± 0.06 C b | 0.81 ± 0.05 D a |
| *Pinus sylvestris* var. *cretacea* (Kalen.) Kom. | Water (c) | 0.40 ± 0.01 A a | 2.23 ± 0.05 B a | 0.99 ± 0.07 C a | 0.73 ± 0.02 D a |
| | Biogor KM | 0.62 ± 0.02 A b | 3.11 ± 0.20 B b | 1.60 ± 0.16 C b | 0.85 ± 0.08 D b |
| | MycoCrop® | 0.57 ± 0.03 A b | 3.14 ± 0.16 B b | 1.53 ± 0.16 C b | 0.88 ± 0.09 D b |

Note: In every cell of the table are presented the average value (first digit) and mean error (second digit). Different capital letters describe statistically significant differences between HMs (in rows). Different lowercase letters describe statistically significant differences between biological preparations Biogor KM and MycoCrop® (in the column). Tukey HSD test, $p < 0.05$; n = 3.

It has been established that the BAC of Cu, Zn, and Cd values are most strongly influenced by the studied factor "biological preparation", with the strength of influence, respectively, $h^2_x(Cu) = 82.9\%$; $h^2_x(Zn) = 78.7\%$; $h^2_x(Cd) = 63.9\%$. There was no significant effect of organized factors on the BAC Pb, which has depended to a greater extent on random factors. A significant effect of the ecotype on the BAC was revealed only for Cu.

**Table 6.** ANOVA two-way analysis of variance of HMs uptake by seedlings of *P. sylvestris* L. on chalk outcrops.

| Resulting Trait | Source of Variation | D | n − 1 | s² | $F_f$ | $F_{0.05}$ | $h^2_x$ |
|---|---|---|---|---|---|---|---|
| BAC Cu | Total | 0.127 | 17 | | | | 100.0 |
| | Reps | 0.001 | 2 | | | | 0.6 |
| | Random | 0.010 | 10 | 0.001 | | | 7.8 |
| | A | 0.005 | 1 | 0.005 | 4.9 | 5 | 3.8 |
| | B | 0.106 | 2 | 0.053 | 53.0 * | 4.1 | 82.9 |
| | A × B | 0.006 | 2 | 0.003 | 3.1 | 4.1 | 4.9 |
| BAC Zn | Total | 3.335 | 17 | | | | 100.0 |
| | Reps | 0.000 | 2 | | | | 0.0 |
| | Random | 0.654 | 10 | 0.065 | | | 19.6 |
| | A | 0.027 | 1 | 0.027 | 0.4 | 8.8 | 0.8 |
| | B | 2.623 | 2 | 1.312 | 20.1 * | 4.1 | 78.7 |
| | A × B | 0.031 | 2 | 0.015 | 0.2 | 8.8 | 0.9 |
| BAC Cd | Total | 1.634 | 17 | | | | 100.0 |
| | Reps | 0.056 | 2 | | | | 3.4 |
| | Random | 0.424 | 10 | 0.042 | | | 25.9 |
| | A | 0.036 | 1 | 0.036 | 0.9 | 8.8 | 2.2 |
| | B | 1.044 | 2 | 0.522 | 12.3 * | 4.1 | 63.9 |
| | A × B | 0.075 | 2 | 0.037 | 0.9 | 8.8 | 4.6 |

**Table 6.** *Cont.*

| Resulting Trait | Source of Variation | D | n − 1 | $s^2$ | $F_f$ | $F_{0.05}$ | $h^2_x$ |
|---|---|---|---|---|---|---|---|
| | Total | 0.198 | 17 | | | | 100.0 |
| | Reps | 0.045 | 2 | | | | 22.6 |
| | Random | 0.085 | 10 | 0.009 | | | 43.0 |
| BAC Pb | A | 0.003 | 1 | 0.003 | 0.3 | 8.8 | 1.4 |
| | B | 0.046 | 2 | 0.023 | 2.7 | 4.1 | 23.4 |
| | A × B | 0.019 | 2 | 0.010 | 1.1 | 4.1 | 9.7 |

Note. Factor A—"pine ecotype"; factor B—"sowing time"; D is the sum of squared deviations (deviant); $s^2$—dispersion; n − 1 is the number of degrees of freedom; $h^2_x$ is the strength of influence on the effective attribute; *—statistically significant differences; n = 3.

Table 7 shows a strong and medium-positive relationship between the accumulation of HMs and the corresponding BAC and nitrification capacity of the soil substrate. Depending on the HM, Spearman's correlation coefficients fluctuated in the absolute values ($r_s$ = 0.610–0.744, $p < 0.05$) and in the value of the BAC ($r_s$ = 0.427–0.765, $p < 0.05$).

**Table 7.** Relationship between the accumulation of HMs, corresponding BAC, and nitrification capacity of the soil substrate.

| Soil Chemical and Biological Indicators | HM Accumulation, mg kg$^{-1}$ | | | | BAC | | | |
|---|---|---|---|---|---|---|---|---|
| | Cu | Zn | Cd | Pb | Cu | Zn | Cd | Pb |
| Nitrification capacity, mg kg$^{-1}$ | 0.744 * | 0.662 * | 0.787 * | 0.610 * | 0.543 * | 0.549 * | 0.765 * | 0.227 |
| N–NO$_3$ content, mg kg$^{-1}$ | 0.132 | 0.006 | 0.067 | −0.222 | −0.031 | -0.109 | −0.087 | −0.293 |
| N$_{total}$ content. % | 0.065 | 0.083 | −0.045 | −0.327 | −0.251 | −0.024 | 0.013 | 0.017 |
| Humus content, % | −0.041 | −0.178 | −0.057 | −0.103 | −0.117 | 0.070 | 0.069 | 0.098 |
| pH | 0.090 | 0.050 | 0.025 | 0.220 | −0.146 | −0.079 | −0.126 | −0.044 |

*—statistically significant correlation; n = 3.

## 4. Discussion

It is necessary to develop methods that provide an understanding of the trends in the genotype-environmental interactions of plant organisms with the environment and, especially, the highly toxic chemicals in the environment. The genetic features of coniferous plants and their diversity make it possible to take into account a wide range of negative factors that limit the favorable growth of forest crops in various environmental conditions.

At present, phytoremediation, i.e., the fixation and neutralization of HMs in plants, is considered to be an environmentally and economically efficient method [78]. In this regard, our approach is consistent with a general global trend in research on the interaction of plants and HMs, which is especially relevant in a hydrographic network of erosion landscapes. The use of various pine ecotypes allows for choosing the right directions in the study of the soil–microorganism–plant system.

In the study, we developed environmentally and economically efficient afforestation measures based on the natural mechanisms of absorption, biotransformation, and bioaccumulation of pollutants in plants [79].

The studies showed that the Scots pine (*P. sylvestris*), including the Cretaceous (*P. sylvestris* var. *cretacea*) one, can become more important for degraded landscapes using the biological preparations Biogor KM and MycoCrop®. This helps to utilize the useful properties of *P. sylvestris*—high polymorphism and high ecological amplitude—for successfully growing in the conditions of the chalk outcrops.

Even though the Scots pine (*P. sylvestris*), as an obligate mycotroph, enters into symbiosis with many ectomycorrhizal fungi species [80,81], the biological preparations Biogor

KM and MycoCrop® improve this capability. The preparations provide fungi with more photosynthesis products, and a host plant mycorrhiza improves plant resistance to poor environments [52,61,82,83].

Reliable data have been obtained showing that mycorrhiza protects pine seedlings from soil pathogens and nematodes, starting from the first stages of plant ontogenesis [84,85]. In this regard, treatment of pine seedlings with spores of ectomycorrhizal fungi increases the survival rate of plants, especially during the first years of life [86,87].

The formation of an ectomycorrhiza and the symbiotic relationships with beneficial microflora are necessary conditions for a pine seedling's survival at the early stages of ontogenesis [81]. In this regard, our data agree with other authors on the need to treat the seeds of *P. sylvestris* with the biological preparations Biogor KM and MycoCrop® based on mycorrhizal fungi. This should become an obligatory method for increasing reforestation's efficiency.

With the use of biological preparations Biogor KM and MycoCrop® based on a consortium of microorganisms during sowing, a greater amount of HMs (Cu, Zn, Cd, and Pb) is accumulated in the Scots pine (*P. sylvestris*) seedlings. This process occurs against a background of a significant increase in the nitrification capacity of the fine aggregate chalk substrate (soil aggregates < 1 mm). Cretaceous pine culture (*P. sylvestris* var. *cretacea*) accumulates a larger amount of HMs compared to *P. sylvestris*, indicating a greater symbiotrophy of *P. sylvestris* var. *cretacea* on carbonate outcrops.

In previous studies conducted on deciduous and coniferous species, it was suggested that a ramified ectotrophic mycorrhiza formation in the soil prevents the entry of HMs into plants [88]. A strong negative correlation was established between the abundance of arbuscular mycorrhizal fungi in the rhizosphere and the HM content in the plants.

However, studies conducted with various coniferous crops have shown that ectomycorrhiza does not have selectivity for trace elements. Simultaneously with the absorption of the necessary amount of mineral substances, the transport and absorption of HMs ions increase [48]. A simultaneous treatment with arbuscular mycorrhiza and microorganisms enhances the effect of HMs absorption and their accumulation in biomass. This increases the stress resistance of host plants to the toxic effects of HMs [89]. These data are consistent with ours: when plants of both *P. sylvestris* and *P. sylvestris* var. *cretacea* ecotypes were treated with the biopreparations Biogor KM and MycoCrop® based on a consortium of microorganisms, along with an increase in the content of HMs in the vegetative mass, survival increased.

An increase in the nitrifying ability of substrates when using biological preparations Biogor KM and MycoCrop®, as well as a high positive relationship between the accumulation of HMs in seedlings and the nitrifying ability of the fine aggregate substrate, confirms the fact of an increase in biological activity in the rhizosphere on a carbonate substrate. Previously, this process has been repeatedly observed in other species—*Galega orientalis* Lam. and *Medicago varia* Mart. [90,91].

There is a close Spearman correlation between the BAC of Cu, Zn, Cd, and Pb in the dry matter of Scots pine (*Pinus sylvestris* L.) seedlings and the nitrification capacity of substrate ($r_s$ = 0.610–0.744, $p < 0.05$), as well as the relationship between the nitrification capacity and the value of the BAC of Cu, Zn, and Cd ($r_s$ = 0.543–0.765, $p < 0.05$). No relationship was found between the BAC of Pb and other indicators.

Improved environmental services for Scots pine (*P. sylvestris*) growth can be achieved using the Biogeosystem Technique (BGT*) methodology [68,92,93]. The BGT* application in the form of intra-soil mechanical processing for soil structure and architecture improvement, intra-soil pulse continuous-discrete watering, and intra-soil matter recycling is capable of enhancing the soil nitrification ability [94], stabilizing and increasing the soil organic matter content [95–97], ensuring the HM intra-soil passivation [98], and providing a long-term efficient nutrition and stimulation function of nanomaterials, polymicrobial biofilms, and humic substances [99–101].

## 5. Conclusions

A result of the study is an improved development of the Scots pine (*P. sylvestris*), including Cretaceous pine (*P. sylvestris* var. *cretacea*) seedlings, treated with the biological preparations Biogor KM, created in Russia on the basis of a consortium of bacteria of the genus Bacillus and mycorrhizal fungi of the genus Glomus, their metabolic products and microelements, and MycoCrop®. These treatments provided a statistically significant increase in the absorption of Cu, Zn, and Cd by Scots pine seedlings, a long-term HMs concentration in the wood, and reliable HMs removal from the soil. We assess this as an important phytoameliorative effect. Biogor KM and MycoCrop® increased the nitrification capacity of the soil at a statistically significant level. The use of Biogor KM enhances Scots pine growth and HMs transfer to a seedling at the same level as MycoCrop®, developed in Germany, provides.

The significance of the work is the possibility of providing reliable reforestation with Scots pine (*P. sylvestris*) and Cretaceous pine (*P. sylvestris* var. *cretacea*) on the chalk outcrops using the biological preparations Biogor KM, MycoCrop®, and BGT* methodology and ensuring soil phytoremediation from HMs.

**Author Contributions:** Conceptualization, V.M.K., A.A.Z., M.G.B. and V.P.K.; data curation, E.V.D., V.G.K., I.V.Z., S.S.M. and M.V.B.; formal analysis, B.B.K., I.V.Z., V.A.C. and S.S.M.; funding acquisition, A.P.G., M.A.S. and L.V.P.; investigation, V.I.C., E.V.D., L.D.S. and V.G.K.; methodology, V.M.K. and V.I.C.; project administration, A.P.G.; resources, A.P.G. and L.V.P.; software, L.D.S., B.B.K. and V.A.C.; supervision, A.A.Z., S.S.M. and V.A.C.; validation, L.L.S., E.V.D., L.R.V. and M.V.B.; visualization, V.I.C., I.V.Z. and M.V.B.; writing—original draft, V.G.K. and V.I.C.; writing—review and editing, M.G.B., M.A.S., L.L.S., L.R.V. and V.P.K. All authors have read and agreed to the published version of the manuscript.

**Funding:** The research was carried out within the framework of the State task on the topic "Immobilization of trace elements by the products of interactions of layered silicates with soil organic matter and microorganisms" (Additional Agreement No. № 073-03-2023-030/2 from 14 February 2023 to Agreement № 073-00030-23-02 from 13 February 2023).

**Data Availability Statement:** Data are available on request to the authors.

**Conflicts of Interest:** The authors declare no conflict of interest.

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
