# Peer review of "Scots Pine (Pinus sylvestris L.) Ecotypes Response to Accumulation of Heavy Metals during Reforestation on Chalk Outcrops"

_forests, doi:10.3390/f14071492_

Round 1

Reviewer 1 Report

Very important topic was chosen for research. In practice, successful reforestation on extreme soils is very difficult. Most of the research so far has only studied the use of different biopreparations for reforestation of extreme terrains. In addition to the reaction of two Scots pine ecotypes to biopreparations, your research covered the accumulation of heavy metals during reforestation on chalk outcrops. You described the method of work clearly and in detail. You have received concrete results that give a concrete answer as to how reforestation can be successfully carried out on land with similar characteristics. I have absolutely no objections to any segment of the research and I must admit that it was a pleasure to review this kind of work.

Author Response

Dear Reviewer 1,

Thank you for your support of our Manuscript. 

Reviewer 2 Report

Manuscript ID: forests-2419852

Title: Scott pine ecotypes response to biological preparations and accumulation of heavy metals during reforestation on chalk outcrops

Article type: Research Paper

Thanks to the authors for the opportunity to read their manuscript. The paper contains interesting and original observations. The paper falls within the Aims and Scope of the journal. The abstract covers the information presented in the manuscript. The paper has an appropriate structure, it is well written and contains up-to-date references. Results are presented in an understandable way. Conclusions are consistent with the evidence. The English typing is adequate. In general, I find this work interesting, and it is giving important information for researchers dealing with heavy metal pollution, and reforestation.

I have number of suggestions to improve the quality of the manuscript detailed below:

1.     At all tables and figures, where it’s needed, you should specify the sample size per group in the text for analysis and give letters or asterisks to indicate statistically significant differences.

2.     Material and methods - this part is where I have the most questions and doubts.

How many seeds were sown in all the plots? How many seedlings grew from the sown seeds? Why accounting plot area was 2m2? How many seedlings were in each study plot? How many biennial seedlings were collected to determine HMs? This important information is missing, and it's difficult to say conclusively whether the experiment was conducted properly.

3.     Abbreviations under table 3 are inappropriate.

4.     What about the significance of the correlation in Table 7? This very important information is missing and needs to be filled in.

5.     Discussion is brief and very cursory.

Author Response

Dear Reviewer 2,

Thank you for your comments to our manuscript.

We assess your comments and proposition as helpful, and accepted all with gratitude.

Our answers and the changes in manuscript are highlighted in red and presented in the comments, a deleted text is on the grey background.

I have number of suggestions to improve the quality of the manuscript detailed below:

  1. At all tables and figures, where it’s needed, you should specify the sample size per group in the text for analysis and give letters or asterisks to indicate statistically significant differences.

The sample size per group in the text for analysis was specified.

The letters and asterisks to indicate statistically significant differences were provided where it’s needed.

  1. Material and methods – this part is where I have the most questions and doubts.

How many seeds were sown in all the plots?

In each plot of experiment, seeds were sown at a rate 100 pcs. viable seeds per 1 m2. Thus, 200 seeds were sown per 1 registration plot (2 m2), and 500 seeds were sown per total plot (5 m2).

How many seedlings grew from the sown seeds?

In the spring of 2019, the number of seedlings, depending on the options for the experiment, varied from 41.5 to 55.8 pcs. per 1 m2 (https://doi.org/10.3390/f14061093).

Why accounting plot area was 2 m2?

The accounting plot area 2 m2 (1m×2 m) has been chosen as optimal for laboratory and field experiments with coniferous crops seedlings based on previous studies (https://doi.org/10.3390/f14061093)

The total area of the plot is 5 m2 (2.5m×2 m). The total area of the gross experiment was 100 m2. The total area of the experimental plots was gross 90 m2. The total net area of the accounting plots was 36 m2.

How many seedlings were in each study plot?

By the autumn of 2020, depending on the variant of the experiment, the number of seedlings in each study plot was from 11.3 to 27.8 pcs./m2.

How many biennial seedlings were collected to determine HMs?

A mixed sample for determining the HM content was formed from 20 biennial seedlings for each repetition of the experiment in three replicates.

This important information is missing, and it's difficult to say conclusively whether the experiment was conducted properly.

We provided all needed information.

  1. Abbreviations under table 3 are inappropriate.

Abbreviations corrected

  1. What about the significance of the correlation in Table 7? This very important information is missing and needs to be filled in.

The significance of correlation marked with *.

  1. Discussion is brief and very cursory.

Discussion has been transformed, new fragments were added including a part of Conclusions section.

Reviewer 3 Report

The authors studied response of two ecotypes of Pinus sylvestris that are growing on chalk soil to heavy metals and effects of different biofertlizers. Generally, article is interesting. Introduction, Methods and Results are adequately desribed. The discussion is not too much connected with the results. Conclusion should be rewriiten. My recommendation: major revision. I gave some remarks, comments and suggestions in the Annotated Manuscript (pdf version), so that the authors can improve this paper. 

Author Response

Dear Reviewer 3,

Thank you for your comments to our manuscript.

We assess your comments and proposition as helpful, and accepted all with gratitude.

Our answers and the changes in manuscript are highlighted in red and reflected in the comments. Deleted text on the grey background.

The discussion is not too much connected with the results.

Higher rate connectivity provided.

Conclusion should be rewritten.

Conclusion were rewritten.

My recommendation: major revision. I gave some remarks, comments and suggestions in the Annotated Manuscript (pdf version), so that the authors can improve this paper. 

All comments and suggestions were accepted with gratitude.

Round 2

Reviewer 2 Report

I accept all corrections and clarifications and recommend the manuscript for publication. 

Author Response

Dear Reviewer 2,

Thank you for your support of our Manuscript ones again.

Reviewer 3 Report

The authors have significantly improved the quality and accuracy of the work. However, it still needs some corrections. My recommendation: minor revision. I gave some comments and suggestions in the Annotated Manuscript (pdf version).

Author Response

Dear Reviewer 3,

Thank you for your new comments to our manuscript.

All your new comments and suggestions we accepted with gratitude again.

Our answers and the changes in manuscript are highlighted in red and reflected in the comments.

Subsection 2.2. Venue and conditions title is changed to the 2.2. Object conditions and experiment layout

We moved all methodological sources to the References

The Table 7. Soil indicators of HMs accumulation, and corresponding BAC title is changed to the Table 7. Relationship between the accumulation of HMs, corresponding BAC and nitrification capacity of the soil substrate.

A paragraph about phytoremediation added to the Introduction section:

The reforestation with the Scots pine (Pinus sylvestris) including Cretaceous pine (Pinus sylvestris var. Cretacea Kalenicz. exCom.) on the chalk outcrops using the biological preparations Biogor KM and MycoCrop® is promising in soil phytoremediation from HMs.

We moved the reference [80 87] one sentence forward.